# Titers of IgG, IgM, and IgA Against SARS-CoV-2 in Healthcare Workers from a General Hospital in Mexico City

**DOI:** 10.3390/diseases13090276

**Published:** 2025-08-22

**Authors:** Nelly Raquel González-Arenas, Mariana Dinazar Chavez-Vargas, Hector Prado-Calleros, Juan Pablo Ramírez-Hinojosa, Fernando Martinez-Hernandez, Angélica Olivo-Díaz, Pablo Maravilla, Mirza Romero-Valdovinos, Guillermina Ávila-Ramírez

**Affiliations:** 1Hospital General “Dr. Manuel Gea González”, SSA. Calzada de Tlalpan 4800, Col Seccion XVI, Tlalpan, Mexico City 14080, Mexico; nelly_raquel@hotmail.com (N.R.G.-A.); hmpradoc@hotmail.com (H.P.-C.); dr.ramirezhinojosa@yahoo.com (J.P.R.-H.); fherxyz@yahoo.com (F.M.-H.); aolivod@yahoo.com (A.O.-D.); maravillap@yahoo.com (P.M.); 2Departamento de Microbiologia y Parasitologia, Facultad de Medicina, Universidad Nacional Autonoma de Mexico (UNAM), Mexico City 04510, Mexico; 517chavez.vargas@gmail.com

**Keywords:** COVID-19, healthcare workers, immunoglobulins

## Abstract

Objectives: The aim of the present study was to better understand the antibody concentrations in healthcare workers (HCWs) from a hospital in Mexico City with a high density of COVID-19 patients. Methods: Up to 243 HCWs were recruited in 2020 and 2022 and were sorted into three groups: hybrid immunity (HI, natural infection plus vaccination), vaccine-induced immunity (VI), and unvaccinated but RT-qPCR negative at the beginning of the pandemic (UV). Peripheral blood and nasopharyngeal swab samples were obtained; additionally, saliva samples were obtained from the UV group. The titers of IgG, IgM, and IgA against the SARS-CoV-2 receptor-binding domain (RBD) and nucleocapsid (NCP) proteins were assessed using an in-house ELISA, and positivity to the virus was determined via RT-qPCR. Results: Most HI and VI participants were positive for serum anti-RBD IgG (92.8% and 100%, respectively), while 26.6% (for HI) and 19% (for VI) were positive for anti-NCP IgG. Regarding serum anti-RBD IgA, the VI and HI groups had positive rates of 87.3% and 66%, respectively. In contrast, the UV group showed a rate of 5.7% but the positivity for IgA in saliva was higher (52% for RBD and 35% for NCP). In addition, the highest antibody titers were obtained for anti-RBD IgG and IgA in the HI and VI groups, respectively. In saliva, the IgA antibody titer was higher for the RBD antigen (1:1280). Conclusions: These results strengthen our understanding of antibody concentrations in HCWs during two critical years of the pandemic in a general hospital with many COVID-19 patients.

## 1. Introduction

Although COVID-19 is no longer defined as a Public Health Emergency of International Concern, it continues to significantly impact global health, particularly in the future, as new variants emerge [1]. More than 460,000 articles about COVID-19, published from 2020 to today, are currently available for download in the PubMed database. However, it is essential to note that each country established different surveillance, treatment, and control strategies at the outset of the pandemic based on its available resources and priorities, which further complicated the viral infection itself, resulting in significant heterogeneity in the pandemic’s final impact at the local level [1].

Healthcare workers (HCWs) played a vital role in fighting the COVID-19 pandemic, making them at greater risk of contracting the disease. Consequently, they were the first groups prioritized for vaccination. A meta-analysis of articles published in 2020 showed that the estimated prevalence of SARS-CoV-2 in HCWs based on RT-PCR was 11% (95% CI: 7–15%) and 7% based on the presence of antibodies (95% CI: 4–11%). Nursing staff had the highest rates of infection (48%; 95% CI: 41–56%), followed by medical personnel working in hospitalization/non-emergency wards during screening (43%; 95% CI, 28–59%) [2]. However, at the local level, these results varied since the prevalence of SARS-CoV-2 infection in HCWs varied according to the population and the methodology used to assess it; e.g., some studies reported a prevalence of 2.4% to 2.7% when detected via PCR [3,4]. A descriptive study performed from March 2020 to mid-2021 on HCWs from a reference center for COVID-19 care in Sao Paulo city, Brazil, reported that 19.12% were positive for anti-SARS-CoV-2 antibodies according to a chemiluminescent immunoassay; interestingly, the resident physicians and healthcare professionals from the institution itself presented the lowest prevalences (nurses, nursing assistants, physicians, and laboratory technicians) [5]. The Centers for Disease Control (CDC) reported that seroprevalence ranged from 0.8 to 31.2% among HCWs in 12 states in the USA [6].

The antibody concentrations in COVID-19 patients vary depending on age, genetics, immunological characteristics, and the time elapsed since infection and vaccination, as well as the type of vaccine (mRNA or vector) and the number of doses administered. A study focused on HCWs with prior SARS-CoV-2 infection who received two doses of an mRNA vaccine against the spike antigen and who provided serum samples longitudinally (separated by at least 90 days, with a serum sample collected at least 14 days after receiving the second dose of vaccine), developed higher spike antibody titers 14 days after receiving the second dose of vaccine compared to vaccinated individuals without prior infection [7]. In another study, seven days after the first immunization with the BNT162b2 mRNA COVID-19 vaccine, HCWs with a previous infection experienced a 126-fold increase in antibody levels; in contrast, a naive HCW group showed a much lower response. After the second dose, no significant increase in antibody levels was found in the previously infected HCWs, whereas in naive HCWs, the levels increased by 16-fold [8]. Therefore, differences in antibody levels cannot be considered a reliable measure of immunity. Furthermore, the observations from monitoring of individuals infected with SARS-CoV-2 suggest a pattern of diminished humoral responses [9,10,11].

Most SARS-CoV-2 vaccines target the spike (S) protein as this virus enters host cells through the interaction between the receptor-binding domain (RBD) of the viral S-protein and the human angiotensin-converting enzyme-2 (ACE2) receptor [12]. Therefore, the production of neutralizing antibodies presents kinetics similar to those of other respiratory viruses [13,14].

In Mexico, vaccines began being administered in December 2020. The first available vaccine was BNT162b2 (BioNTech-Pfizer, New York, NY, USA), which is administered via intramuscular injection, with subsequent boosters using the same vaccine or other available ones, such as mRNA-1273 (Moderna, Cambridge, MA, USA), ChAdOx1-S (Oxford-AstraZeneca, Oxford, UK), CoronaVac (Sinovac, Beijing, China), Convidecia (CanSino, Tianjin, China), or Sputnik V (Gamaleya, Moscow, Russia) [15]; the last three became locally available beginning in mid-2021.

The objective of the present study was to compare the systemic and mucosal humoral immune responses to SARS-CoV-2 in healthcare workers (HCWs: residents, medical doctors, and nurses) immunized by vaccination, natural occupational viral exposure, and hybrid immunization (vaccination and natural occupation-related viral infection).

## 2. Materials and Methods

### 2.1. Samples

HCWs at the Hospital General “Dr. Manuel Gea Gonzalez” (HGMGG) in Mexico City were invited to participate in the present study in 2020 and 2022. The participants were medical staff of the infectious diseases, anesthesiology, otorhinolaryngology, and surgery departments (residents, medical doctors, and nurses), and were divided into three groups according to the sampling time and their follow-up characteristics: a hybrid immunity group (HI), which was recruited from January to December of 2022 and consisted of those who were positive for SARS-CoV-2 based on RT-qPCR, had or did not have respiratory symptoms at the time of sampling, and who reported having been previously vaccinated. The vaccine-induced immunity group (VI) was enrolled from January to December of 2022 and included participants who reported no previous COVID-19 infection, were SARS-CoV-2 negative based on RT-qPCR, and had received at least one dose of a SARS-CoV-2 vaccine. Finally, the unvaccinated group (UV) consisted of participants who reported no previous COVID-19 infection, were SARS-CoV-2 negative based on RT-qPCR at the time of sampling, and had not been vaccinated (since they were recruited from June to December 2020, when vaccines were not yet available).

The Ethics and Research Committees of HGMGG approved the study under reference numbers 12-26-2020 and 12-111-2020. Written informed consent was obtained from all participants before their recruitment into the study.

Nasopharyngeal swabs and 5 mL peripheral blood samples were collected from all participants. In addition, 3 mL saliva samples were obtained from the UV group. Serum, saliva, and swab fluids were recovered and inactivated by heating at 56 °C for 30 min. They were then divided into aliquots and stored at −70 °C until the analysis. A questionnaire about clinical data such as age, sex, comorbidities, weight, SARS-CoV-2 vaccination schedule, and, for symptomatic participants, peripheral blood laboratory data and biochemistry profiles related to COVID-19.

### 2.2. RNA Extraction and SARS-CoV-2 Detection via qRT-PCR

RNA extraction from samples was performed using an automated nucleic acid platform (Maelstrom 9600, TANBead, Taipei, Taiwan) with a commercial extraction system (TANBead Viral Nucleic Acid Extraction Kit, Tiangen, Beijing, China), and SARS-CoV-2 detection was performed using the SARS-CoV-2 Real-Time PCR Detection Kit (Open Reading Frame (ORF1ab) and Nucleocapsid (N) genes) (Viasure, CerTest Biotec, Zaragoza, Spain), following the manufacturer’s instructions.

### 2.3. Evaluation of Anti-SARS-CoV-2 Antibodies via Enzyme-Linked Immunoassay (ELISA)

IgM, IgA, and IgG immunoglobulins against recombinant proteins for the Nucleocapsid (NCP, Sino Biological 40592-V08B) and Spike-RBD of the SARS-CoV-2 virus (Sino Biological 40592-V08B) were detected in the serum and saliva samples [16]. An indirect in-house ELISA system was used [13]: 96-well plates (Immulon 2HB, flat bottom 3455 Thermo, Waltham, MA, USA) were coated with 100 μL of the NCP or Spike-RBD antigen at 0.5 μg/mL in PBS and left at 4 °C overnight. The plates were blocked with a commercial solution (P-01-37515 Thermo) for one hour, and then 100 μL of serum was added at a dilution of 1:40. After 2 h of incubation at 37 °C, the plates were washed, and the corresponding peroxidized secondary antibody was added and the plate was incubated again at 37 °C for 1 h. The plates were then washed, 100 μL of Sigmafast OPD substrate (Sigma Aldrich St. Louis, MA, USA) was added, and the reaction was incubated for 5 min. The reaction was stopped by adding 50 μL of a 2N H_2_SO_4_ solution. Absorbance values were obtained at 490 nm using an ELISA reader (iMark Bio-Rad, Hercules, CA, USA). The ELISA test for saliva samples was performed as described for serum, except that the saliva samples were diluted 1:20 and anti-human IgA was employed at a dilution of 1:2000. The titer of the samples in which SARS-CoV-2 antibodies were detected was determined following the ELISA protocol mentioned above and using serial dilutions (1:40; 1:640; 1:2560; 1:10,240, and 1:40,960 for serum; 1:20, 1:40, 1:80, 1:160, and 1:320 for saliva) according to the method of Pattinson et al. [17].

To determine the validity of the ELISA test, 40 serum and saliva samples from individuals who had not had COVID-19, collected during the 2009 H1N1 influenza pandemic, were used as negative controls, and 37 samples from individuals with previously confirmed COVID-19 (displayed symptoms and positive RT-qPCR test) were used as positive controls. The optimal cut-off points for sample positivity were defined as the mean optical density of the negative controls, considering two standard deviations. The results were expressed as the ELISA index (EI), where EI = sample absorbance/cut-off absorbance. EI values >1 were considered positive. The specificity and sensitivity for the detection of IgM, IgA, and IgG were also determined.

### 2.4. Statistical Analysis

Graphs were generated and statistical analysis was performed using GraphPad Prism v10.2.3 (GraphPad Software, San Diego, CA, USA). Normality analysis was conducted using the D’Agostino–Pearson omnibus test for EI and Levene’s test for the clinical and general patient data. A non-normal distribution in the IE values was observed. The ELISA results were expressed as the medians and interquartile ranges, and multiple comparisons were performed with the non-parametric Dunn’s test if the Kruskal–Wallis test indicated general differences in the median IE values of the different immunoglobulin titers. The association between antibody titers and the presence of symptoms, as well as the differences in the clinical and general characteristics of the groups, was assessed using ANOVA for continuous data and the two-tailed Pearson X^2^ test or Fisher’s test as appropriate for categorical data, with *p* ≤ 0.05 indicating statistical significance. Fisher’s exact test was used when the expected frequency in at least one cell was 5 or less. Odds ratios (ORs) and 95% confidence intervals (95% CIs) were calculated using the Cornfield approximation in IBM SPSS Statistics for Windows Version 27 (IBM Corp., Armonk, NY, USA).

## 3. Results

A total of 243 participants were recruited and classified according to their characteristics; 112, 63, and 68 were assigned to the HI, VI, and UV groups, respectively. Although all participants answered most of the questionnaire questions on clinical variables, responses to some items were missing, mainly due to the accuracy of dates. The clinical characteristics of the participants are shown in Table 1. On average, the age of participants was ~40 years, most were overweight or obese, the most frequent symptom was cough for the HI group (63.4%), headache for the VI group (62%), and rhinorrhea for the UV group (8.8%). Additionally, the majority of participants were vaccinated with Pfizer (69.6% of the HI group and 66.7% of the VI group), followed by Astra-Zeneca (22.3% of the HI group and 25.4% of the VI group), and more than 93% of the HI and HV groups received two vaccine doses. Statistical analysis of some clinical and general characteristics showed significant differences between groups, particularly in comparisons of weight and some respiratory symptoms.

For serum samples, the ELISA for anti-NCP IgG showed the highest sensitivity (97.4%); the sensitivity of the anti-RBD IgG ELISA was also high (97.2%). The performance of the IgA detection assay showed 89.5% and 100% sensitivity for NCP and RBD, respectively. While the IgM assay showed lower sensitivity for both antigens, the sensitivity for anti-NCP IgM was 68.4% and for anti-RBD IgM, it was 80.6%. All the assays were 100% specific. The assays developed for IgA in saliva were 100% sensitive and specific for both antigens. The precision of the standardized assays was measured. Different serum or saliva control samples were assayed in replicates on separate days. The precision is given as the standard deviation and coefficient of variation (Appendix A). The performances are in line with those in a previously published paper [18].

The validated ELISAs were then used to measure the different antibody isotypes. Table 2 summarizes the results for the IgM, IgA, and IgG antibody responses against Spike-RBD and NCP in the serum and saliva samples from the HCWs (saliva samples were only assessed in the UV group). Among the HI and VI participants, 33.9% and 41.3% were positive for anti-RBD IgM; in contrast, the response to NCP of these groups was low: 21.3% were positive in the HI group and 9.5% in the VI group. Regarding IgA antibodies, 66% and 87.3% tested positive for Spike-RBD in the HI and VI groups. For NCP, only 23.2% and 3.2% were positive in the HI and VI groups. Furthermore, the anti-RBD IgG response was the highest (Figure 1A), as 100% and 92.8% of the participants in the VI and HI groups exhibited a response, with statistical differences between groups. In contrast, the anti-NCP IgG response was lower, showing positivity in 19% and 26.8% of the VI and HI groups, respectively. Interestingly, the UV group presented the lowest systemic response, with statistical differences compared to the other groups (Figure 1B). In contrast, the mucosal response in the UV group was higher, with anti-RBD and NCP IgA positivity rates in saliva of 52.9% and 35.7%, respectively.

Regarding antibody titers, the IgM titers against both antigens were generally low, especially in the HI and UV groups, with lower titers than the initial working dilution (1:40); even the EI values were close to the cutoff. Only participants in the VI group reached titers of 1:640 for anti-RBD and anti-NCP IgM (Figure 2A). Regarding serum IgA antibody titers, high IgA titers were observed in the HI group (1:640 dilution) for both antigens. In the VI group, only IgA for the RBD antigen reached titers of 1:640; similarly, in the UV group, only IgA against the NCP antigen reached titers of 1:640. The EI values were high in the samples at the initial dilution but declined rapidly at subsequent dilutions (Figure 2B). Regarding IgG titers, the HI and VI groups reached titers of 1:2560 and 1:10,240, respectively, for the RBD antigen, while the NCP IgA titers were 1:40. In contrast, in the UV group, a higher titer of anti-NCP IgG antibodies (1:640) was observed compared to anti-RBD IgG antibodies (1:40) (Figure 2C). It can be observed that for the HI and VI groups, the humoral response against the virus was more intense in comparison with the UV group. The IgA titers in the UV group were lower for NCP in saliva compared to serum, and higher titers of IgA against the RBD antigen were observed in saliva (1:1280) than in serum (1:40) (Figure 2B,D). No significant association was found between antibody titer and clinical and laboratory variables (Appendix A).

## 4. Discussion

We evaluated the humoral immune response against the SARS-CoV-2 virus in healthcare workers at a Mexican public hospital. The seropositivity rates observed in the HI group were due to the effects of both the SARS-CoV-2 vaccine and ongoing SARS-CoV-2 infections; therefore, in the ELISA tests, a stronger humoral immune response was detected for the Spike-RBD antigen than for NCP (consistent with their mild disease symptoms) [19]. Including the VI group allowed for an assessment of the immunity induced solely by vaccination, as all participants presented anti-Spike-RBD IgG, and a greater number were positive for IgM and IgA against Spike-RBD. However, antibodies against NCP were also identified despite negative RT-qPCR results and no prior SARS-CoV-2 infection. This could be explained by the presence of asymptomatic infections that went unnoticed, confirming the effectiveness of vaccination in this occupational risk group [19]. Furthermore, although cross-reaction with seasonal coronavirus cannot be ruled out, a cohort study conducted between 2010 and 2014 in six hospitals in Mexico looking for respiratory pathogens, including coronaviruses NL63, OC43, and 229E, only detected these coronaviruses in very few cases (frequency of 0.6%) and they were associated with respiratory syncytial virus infection [20,21], so the possibility of a cross-reaction with seasonal coronavirus in our study is too low.

Antibody production against NCP, especially IgG antibodies, is common in patients recovering from acute SARS-CoV-2 infection. It has been shown that during the convalescent phase of a SARS-CoV-2 infection (up to 100 days from symptom onset), 89.8% of patients produced anti-NCP IgG, while 92.9% produced a combination of IgG and IgM [22].

IgG against the SARS-CoV-2 NCP antigen has been proposed to be a more robust and reliable measure of acquired immunity against the virus than IgM, as the latter decreases at a faster rate [9]. Our results are in agreement with this since we found higher levels of anti-NCP IgG production compared to IgM, with positive titers that were maintained during the serial dilutions.

On the other hand, the UV group was particularly interesting, since including this group allowed us to study the humoral response to exposure to the virus. Overall, this group showed a weak humoral immune response, where only anti-NCP IgG reached 10% positivity. Our results for anti-NCP IgG antibodies (10%) are lower than those reported by Basto-Abreu et al. [23] and Herrera-Ortiz et al. [24], who found a prevalence close to 20% for anti-NCP IgG. The lower positivity rate found in our study may be due to the fact that the UV group did not report a prior SARS-CoV-2 infection, while in the Herrera-Ortiz study, 13.1% reported a history of a positive RT-qPCR result, which may have contributed to the higher positivity rate. A similar finding occurred in the Basto-Abreu study, where they reported a positivity rate of 19.6% but included symptomatic or mildly symptomatic individuals [23,24]. Furthermore, in the same study, 13.6% of the unvaccinated participants had antibodies against the S-protein or NCP; similar results were obtained in the UV group against both antigens in our study.

Although serological studies are commonly performed using ELISA and a single serum dilution, requiring only a small sample volume, this could introduce bias as the measurement resolution is potentially affected if the dilution used produces readings close to the minimum or maximum optical density (OD) of the assay. Therefore, it is common practice to measure OD in a series of 8 to 12 dilutions and report the last dilution, known as the “endpoint at which the OD falls below a predefined baseline value.” However, performing ELISA tests with so many serial dilutions per sample is labor- and resource-intensive. Alternatively, some commercial ELISA kits include a reference serum or standard control from which a calibration curve is constructed; arbitrary units (AU)/mL values are then obtained by extrapolating the sample’s OD value. In addition, one study showed that using a Bayesian hierarchical model, the results from a few dilutions were similar to those using several serial dilutions [17].

In the present study, we evaluated the antibody titers through in-house ELISAs using the dilutions used by Pattinson et al. [17]. The early ELISAs used for COVID-19 diagnosis showed that most sera from convalescent patients were IgG positive at dilutions of 1:40 and were considered “truly” recovered from COVID-19 [19]. Early administration of high-titer SARS-CoV-2 convalescent plasma (S-protein IgG titers > 1:1000) to older adults with mild infection has been shown to reduce the progression of COVID-19 [25].

One study assessed the immunoglobulin (Ig) response to different SARS-CoV-2 antigens using plasma and saliva samples collected from vaccinated subjects (after a second dose of the Pfizer-BioNTech or Moderna vaccine) and convalescent subjects. All the subjects exhibited positivity for total plasma anti-RBD Igs at dilutions of 1:6400 and 1:102,400, while in saliva, dilutions of 1:4 and 1:16 were positive [26]. In addition, positivity for total plasma antibodies against NCP was observed in all participants in the convalescent group at a dilution of 1:6400, but not in vaccinated subjects; meanwhile, the saliva samples from this group showed positivity at 1:2 and 1:16 dilutions. Interestingly, only one vaccinated participant was positive for plasma Igs against NCP at a dilution of 1:100 [26]. In the present study, all the participants showed detectable serum IgG, IgM, and IgA titers at dilutions of at least 1:40, with some participants in the VI group achieving titers of 1:10,240 for anti-RBD IgG.

Most samples showed a marked decrease in serum IgG, IgM, and IgA concentrations between the 1:40 and 1:640 dilutions; however, at the next dilution of 1:2560, the majority of the samples became negative. A comparison of IgG and IgA serum concentrations showed that they exhibited similar kinetics toward RBD and NCP antigens. The antibody titer declines, and the concentrations at different serum dilutions are consistent with previously published data [26]. Furthermore, we found no association between antibody levels and the development of symptoms in the participants in the HI and VI groups. This could be due to the mild nature of COVID-19 and the fact that the participants already had immunity induced by vaccination or by a previous infection and subsequent vaccination.

Interestingly, for most of the saliva samples from the UV group, there was a higher concentration of IgA antibodies against RBD compared to NCP. Thus, most samples were positive up to dilutions of 1:640 and 1:80 for RBD and NCP, respectively. Our results are consistent with previously published data, which indicate that antibody titers are lower in saliva than in serum in COVID-19. A systematic review showed that salivary IgA identification is a suitable alternative for population monitoring instead of using blood samples [27]. In addition, it has been argued that the role of mucosal immunity in COVID-19 is significant, directing both local and systemic immune responses to the virus; however, there are many publications that focused on antibodies and systemic cellular immunity, while there are fewer studies on the mucosal immune response in the respiratory tract, which plays a key role in the early restriction of viral replication and elimination of SARS-CoV-2 [28].

Some limitations of our study were as follows: (i) at the time of the sample collection, time had elapsed since the onset of symptoms and vaccination, which could have had a significant impact on the antibody concentrations and the degree of neutralizing activity present [7,8,9,10]. In this case, we were unable to establish the timing of participants’ development of infection and symptoms. This was primarily due to the lack of interest shown by HCWs when they were responding to the questionnaire, which could have been motivated by burnout syndrome and uncertainty about their employment status during the study’s sampling period, as documented in other studies [29,30]. (ii) A second problem was that we were unable to determine which viral variants infected the participants. In addition, several different vaccines were administered to the participants, as at least four different vaccines (BNT162b2, ChAdOx1-S, Convidecia, and Sputnik V) were available for administration at the time of the study. (iii) There was a potential selection bias and a limited sample size since the HCWs were recruited from a few hospital departments, and they do not represent the total HCW population (n = 1450). (iv) Finally, the descriptive cross-sectional design used in the present study in itself presents limitations for performing various analyses and inferences of causality, while a longitudinal follow-up study would have allowed for more in-depth analyses.

Despite these limitations, our results are consistent with those of other studies on different populations regarding the dynamics and concentration of serum and salivary antibodies, strengthening our knowledge about antibodies in HCWs during two critical years of the pandemic in a general hospital in Mexico with a high concentration of COVID-19 patients.

## 5. Conclusions

In general, most vaccinated and naturally infected individuals were found to have IgG antibodies against RBD at a titer of 1:40, which decreased rapidly after dilution, while a quarter were positive for IgG against NCP. Interestingly, while the unvaccinated group showed a weak serum antibody response against both antigens, the positivity rate for IgA antibodies in saliva was higher—an interesting finding regarding the role of mucosal immunity at the beginning of the COVID-19 pandemic. Therefore, our results are in accordance with previous data regarding the antibody responses in individuals who had recovered from COVID-19 following vaccination and in those who were unvaccinated, particularly focusing on the case of a general hospital with a high concentration of COVID-19 patients during two critical years of the pandemic. Furthermore, the epidemiological overview of seroprevalence and antibody titers is valuable in providing insight into the dynamics of the COVID-19 epidemic, particularly in groups occupationally exposed to the virus, such as healthcare workers.

## Figures and Tables

**Figure 1 diseases-13-00276-f001:**
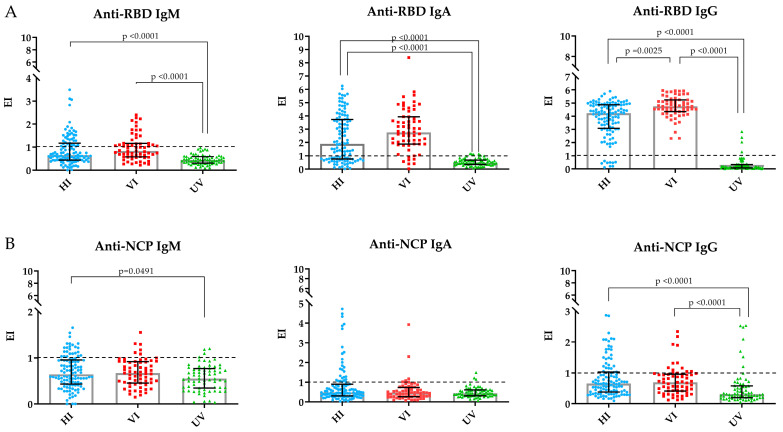
Dunn’s multiple comparisons test. Plots show the statistically significant differences in the median seropositivity of the three types of antibodies against the RBD (**A**) and NCP (**B**) antigens in the HI, VI, and UV groups. The baseline value is indicated with a dashed line, and statistically significant *p*-values are shown.

**Figure 2 diseases-13-00276-f002:**
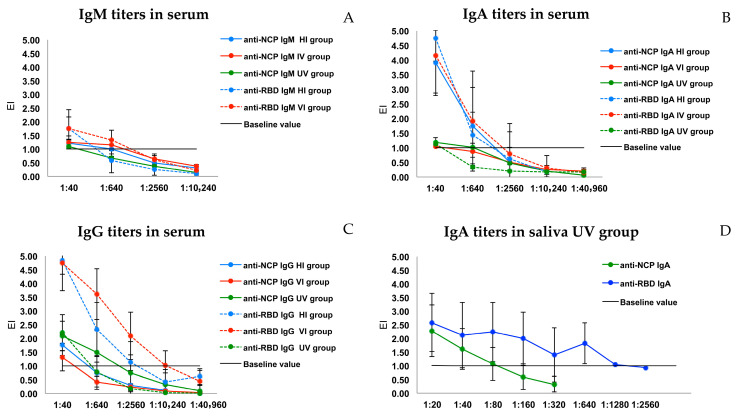
Titers of antibodies against RBD and NCP antigens in the three groups. (**A**) IgM titers in serum. (**B**) IgA titers in serum. (**C**) IgG titers in serum. (**D**) IgA titers in saliva.

**Table 1 diseases-13-00276-t001:** Clinical and general characteristics of the participants.

	HI Group (n = 112)	HI vs. VI (*P*) ^c^	VI Group (n = 63)	HI vs. UV (*P*)	UV Group (n = 68)	VI vs. UV (*P*)
Average age ± SD ^a^, years	40.25 ± 14.23	0.701	39.1 + 13.9	0.979	41.8 + 12.8	0.646
Female/male, n	61/51	0.453	38/25	0.444	41/27	0.998
Body mass index (mean± SD)	26.52 ± 5.10	0.984	26.51 ± 4.48	0.039	25 ± 0.89	0.043
**Overweight or Obesity ^b^, n (%)**						
Low weight/Normal weight vs. Overweight/obese	50 (45.9)	0.069	20 (31.7)	0.158	37 (56.9)	0.004
	59 (54.1)		43 (68.3)		28 (43.1)	
Normal weight vs. Overweight only	50 (56.8)	0.036	20 (38.5)	0.640	37 (60.7)	0.014
	38 (43.2)		32 (61.5)		24 (39.3)	
Normal weight vs. Obese	50 (70.4)	0.554	20 (64.5)	0.031	37 (77.0)	0.016
	21 (29.6)		11 (35.5)		5 (11.9)	
**Comorbidities, n (%)**						
Diabetes	18 (16)	0.389	7 (11.1)	0.007	2 (2.9)	0.061
Hypertension	19 (16.9)	0.674	9 (14.2)	0.004	2 (2.9)	0.018
Cardiovascular diseases	6 (5.4)	0.222	1 (1.6)	0.052	0 (0)	0.297
**Main symptoms, n (%)**						
Cough	71 (63.4)	0.057	31 (49.2)	<0.001	2 (2.9)	<0.001
Fever	27 (24.11)	0.245	10 (15.9)	0.001	3 (4.4)	0.026
Myalgia	45 (40.2)	0.761	25 (39.7)	<0.001	4 (5.9)	<0.001
Anosmia	10 (8.9)	0.055	1 (1.6)	0.459	4 (5.9)	0.200
Rhinorrhea	23 (20.5)	0.098	20 (32.0)	0.038	6 (8.8)	0.001
Diarrhea	10 (8.9)	0.626	4 (6.3)	0.043	1 (1.5)	0.125
Fatigue	50 (44.6)	0.278	33 (52.4)	<0.001	5 (7.4)	<0.001
Odynophagia	9 (8)	0.498	7 (11.1)	0.588	4 (5.9)	0.281
Headache	67 (59.8)	0.608	39 (62.0)	<0.001	3 (4.4)	<0.001
Dysgeusia	6 (5.4)	0.465	5 (7.9)	0.446	2 (2.9)	0.188
**SARS-CoV-2 RT-qPCR result**						
Negative	0 (0)	<0.001	63 (100)	<0.001	68 (100)	NA
Positive	112 (100)		0 (0)		0 (0)	
Vaccinated n (%)	112 (100)	NA	63 (100)	NA	0 (0)	NA
**Vaccine, n (%)**						
Pfizer	78 (69.6)	0.921	42 (66.7)	NA		NA
Cansino	2 (1.8)		1 (1.6)			
Sputnik	7 (6.3)		4 (6.3)			
Astra Zeneca	25 (22.3)		16 (25.4)			
**Doses, n (%)**						
One dose	7 (6.3)	0.377	2 (3.2)	NA		NA
Two doses	105 (93.7)		61 (96.8)			
**Time elapsed from the last dose to sampling, n (%)**						
14–30 days	7 (6.3)	0.043	0 (0)	NA		NA
31–90 days	12 (10.7)		3 (4.8)			
>90 days	93 (83)		60 (95.2)			

^a^ SD: Standard deviation. ^b^ Based on standard interpretations of body mass index (BMI) values: > 18.5 = low weight; 18.5–24.9 = normal weight; 25.0–29.9 = overweight; >30.0 = obesity. ^c^ *P*: Probability. NA: Not applicable.

**Table 2 diseases-13-00276-t002:** Positivity and titers of IgM, IgA, and IgG in serum and saliva samples.

	HI Group			VI Group			UV Group		
	n (%)	Median EI [IQR]	ELISA titer	n (%)	Median EI [IQR]	ELISA Titer	n (%)	Median EI [IQR]	ELISA titer
**Systemic** **Immunity**									
Anti-RBD IgM	38 (33.9)	1.417 [1.144–1.709]	1:40	26 (41.3)	1.366 [1.092–1.797]	1:640	0 (0)	-	-
Anti-RBD IgA	74 (66.0)	3.330 [1.903–4.308]	1:640	55 (87.3)	3.064 [2.168–4.350]	1:640	4 (5.7)	1.126 [1.055–1.161]	1:40
Anti-RBD IgG	104 (92.8)	4.293 [3.336–4.905]	1:2560	63 (100)	4.750 [4.346–5.239]	1:10,240	4 (5.7)	2.214 [1.475–2.724]	1:40
Anti-NCP IgM	24 (21.3)	1.198 [1.091–1.306]	1:40	6 (9.5)	1.230 [1.083–1.365]	1:640	4 (5.7)	1.136 [1.058–1.247]	1:40
Anti-NCP IgA	26 (23.2)	1.720 [1.207–3.706]	1:640	2 (3.2)	1.046 [1.016–1.076]	1:40	4 (5.7)	1.356 [1.144–1.497]	1:640
Anti-NCP IgG	30 (26.8)	1.545 [1.203–2.041]	1:40	12 (19)	1.241 [1.062–1.916]	1:40	7 (10)	2.093 [1.525–2.525]	1:640
**Mucosal** **Immunity**									
Anti-RBD IgA	-	-	-	-	-	-	37 (52.9)	1.909 [1.385–2.319]	1:1280
Anti-NCP IgA	-	-	-	-	-	-	25 (35.7)	1.583 [1.276–2.235]	1:80

EI = ELISA Index; IQR = interquartile range.

## Data Availability

All relevant data are contained within the article.

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
