# Peer review of "Titers of IgG, IgM, and IgA Against SARS-CoV-2 in Healthcare Workers from a General Hospital in Mexico City"

_diseases, 2025, doi:10.3390/diseases13090276_

Round 1
Reviewer 1 Report
Comments and Suggestions for Authors
The present study by Gonzalez-Arenas and Colleagues describes the humoral immune response in HCWs from a general hospital in Mexico City. This study strengthens the understanding of antibody concentrations in HCW during two crucial years of the pandemic in a hospital with a high concentration of COVID-19 patients. However the manuscript requires „Major Revisions“ because of substantial weaknesses in content clarity, grammar issues, methodological transparency, and the lack of practical implications.
Below the authors will find some suggestions and question regarding the manuscript: (see attached file)

The authors have applied the methodology properly (although in some parts it lacks transparency), but the text contains some shortcomings that limit the quality of the manuscript. I therefore recommend that the text be revised with regard to grammatical aspects, style and comprehensibility
Author Response
Reviewer 1
We deeply appreciate your comments and suggestions, which have greatly helped to improve our manuscript. Thank you very much.
The abstract is missing the objective of this study, e.g. the aim of this study was to better understand the antibody concentrations in HCW in a hospital with a high density of COVID-19 patients…??
R: The abstract has been changed and your suggestion was included (lines 16-18).
As various groups (HI, VI, NE) and antibodies (IgG, IgA, IgM) are described here, it can quickly become unclear. I suggest that the authors define a clear and uniform structure for the description of the events, specify all relevant values and avoid repetitions, see examples below:
Since the reader does not yet know how many individuals are in the HI, NE and VI group, absolute or relative frequencies should be provided at this point, e.g. „ Most HI and VI participants were positive in serum to IgG against RBD, and about a quarter were positive to IgG against NCP“. „Concerning the IgA antibody titer in saliva, a higher value was obtained with the RBD antigen“. „ The NE group developed a low antibody response against both antigens.“
R: Thank you, we included the following changes: “Most HI and VI participants were positive in serum to IgG anti-RBD (92.8% and 100%, respectively) while than 26.6% (for HI) and 19% (for VI) were positive to IgG anti-NCP.” (lines 26-27)
„Regarding the IgA vs RBD response, group VI had a positive rate of 87.3% and group HI 66%. …. The NE group developed a low an …“ When describing the groups, please use a in the entire manuscript, for example: VI group
R: Thank you, the manuscript has been carefully reviewed for consistent spelling.
Here the reader have no comparison to NE group for serum antibodies response: „The NE group developed a low (???) antibody response against both antigens. In contrast, in the same group, the positivity for IgA antibodies recorded in saliva was higher (52% for RBD and 35% for NCP).“
R: In the new version of our manuscript, the sentence has been clarified: “… the UV group showed a rate of 5.7% but the positivity for IgA in saliva was higher (52% for RBD and 35% for NCP).” (lines 29-30).
Repetition to the sentence in line 24-25
R: Thank you, the text has been changed.
What do you mean by „pandemic final impact“ – please specify this statement
R: The sentence has been deleted.
47-51 very long sentence with many grammar mistakes. Please shorten the sentence and check the grammar/clarity, e.g.: Begging=beginning there was not a vaccine available = there was no vaccine available transfusion of immunized serum = convalescent plasma? Was assayed = was tested
R: We apologize for the mistakes, the text has been changed and your suggestions were added.
The information provided here is insufficient. The antibody concentration also depends on the time elapsed since vaccination, as well as the type of vaccine (mRNA or vector) and the number of doses administered.
R: Thank you, your suggestion has been included (lines 66-67).
Please check and rewrite this sentence: „Therefore, differences in antibody levels do not allow them to be considered a consistent measure of immunity“ do not allow them = allow whom? (not clear and precise spelling) Consistent meassure= reliable measure
R: Thank you again, the text has been checked and your suggestions were added (lines 77-79).
This sentence lacks consitsency, clarity and has grammar errors, e.g. HCWs play a vital role in fighting the disease, but they do not play a leading role in controlling the pandemic. This statement is misleading. Then=Consequently
were the priority group to receive it = were among the first groups prioritised for vaccination
R: We are sorry, the text has been checked and errors were corrected (lines 47-54).
It remains unclear whether only (doctors=physicians) were the first to receive these vaccinations or whether nursing staff (who also have close contact with patients) were included as well. This is not clearly stated in the current wording.
Moreover, although various vaccines are listed in lines 69–70, not all of them were available as booster doses at the beginning of 2021. This may be misleading and should be clarified.
R: The paragraph was modified and your concerns were clarified (lines 85-90).
The introduction would benefit from a more comprehensive view on vaccine effectiveness over time. there is no synthesis of how protection wanes after vaccination (depending on the number of vaccines inkl. booster), nor how this varies across different available vaccines (mRNA or vector vaccines). The role of antibody titers as a potential correlate of vaccine-induced immunity and protection against severe COVID-19 is not really addressed. That information would help interpret immune response data provided in this manuscript in the result part.
The implications of hybrid immunity are completely missing. Evidence suggests that individuals with prior SARS-CoV-2 infection who subsequently received mRNA vaccines developed significantly higher and more sustained antibody responses compared to those who were only vaccinated. A brief summary of relevant findings would strengthen the introduction. (e.g., authors could cite the below suggested publications or more recent publications on this topic:
Zhong et al., 2021 (DOI: 10.1001/jama.2021.19996);
Ontanon et al. 2021 ( DOI: 10.1016/j.ebiom.2021.103656
or Gray et al. 2021 ( doi: https://doi.org/10.1101/2021.12.28.21268436
R: Once again, we deeply appreciate your recommendations. The Introduction has been substantially revised based on your suggestions, and the recommended references were added (references 7 and 8).
How were the participants recruited? Were the tests conducted as part of routine procedures or for study-related purposes? From which hospital departments or units were the participants recruited? How much time has passed on average since the infection or vaccination? In addition, the vaccination with which the participants were vaccinated is not described. Furthermore, no distinction is made as to how many vaccine doses the I and VI groups had. However, all this information is important in order to be able to classify the results!
R: In the revised manuscript, the recruitment of participants has been detailed (lines 98-101) and information about on the time elapsed for vaccination, the vaccine, and dose received has been added (Table 1).
Check the grammar and shorten the sentence, e.g.:reported never having COVID-19 = having had COVID-1
R: Ok, the text has been checked.
„…12-111-2020, Written informed consent…“ → use period insted of comma
Putting the references in the middle of the sentence makes it bit confusing to read. Please cite at the end of the sentence if appropritate: „… performed [13]: 96-well plates“
R: Ok, the changes have been made.
The statistics part lacks a bit clarity, e.g. Normality analysis was achieved = the distribution of metric (or continuous) data was assessed… „…a non-normal distribution was observed“ = that belongs to a result part If the authors report medians, they should also report interquartile ranges Please provide justification for the use of Dunn's test (according to which global test? e.g. Kruskal-Wallis-Test?). Were all Pearson’s χ² tests two-tailed? Beginn new sentence: A minimum significance level of p ≤ 0.05 was used
R: The statistics section has been extended for its clarification. Now it reads: “Normality analysis was achieved using D´Agostino-Pearson omnibus test for EI and Levene's test for clinical and general patient data were performed. A non-normal distribu-tion in the IE values was observed. ELISA results were expressed as the medians and in-terquartile ranges, and multiple comparisons were performed with the non-parametric Dunn's test, after the Kruskal-Wallis test indicated general differences in the median IE values of the different immunoglobulins. The association between antibody titers and the presence of symptoms, as well as the differences in the clinical and general characteristics of the groups was assessed using ANOVA for continuous data and two tailed Pearson X2 test or Fisher’s test as appropriate for categorical data, with a minimum significance level of p≤0.05. Fisher’s exact test was used when the expected frequency in at least one cell was 5 or less. Odds ratio (OR) and 95% confidence intervals (95% CI) were calculated using the Cornfield approximation in IBM SPSS Statistics for Windows Version 27 (Armonk, NY, USA: IBM Corp).”
In the methods section it is described that comorbidities (line 68) were also recorded, but they are not specified in the description of the study population, why? In addition, the job title of the persons recorded is missing here
R: Table 1 has been modified and new information about comorbidities, vaccines, and doses was added.
Please correct: > 8.5 = < 18.5
R: The mistake has been corrected.
Sensitivity or Specifity % is missing for IgA: „… 100% for IgA,…“
R: Thank you, the missing data have been added, now it reads: “The performance of IgA detection showed 89.5% and 100% sensitivity for NCP and RBD, respectively.” (lines 199-200)
Sensitivity or Specifity % is missing for IgG: „… 100% for IgG,…“
R: Ok, now it reads: “In serum, anti-NCP IgG showed the highest sensitivity (97.4%), and anti-RBD IgG was also high (97.2%).” (lines 198-199)
Provide the % positive in the VI group for IgG response
R: Ok, we have implmented your suggestion.
Check the grammar (comma placement); remove „Interestingly“ → interpretative
R: Ok, now it reads: “In contrast the IgG response anti-NCP was lower, showing positivity in 19% and 26.8% for VI and HI groups, respectively”.
Certainly the word “Interestingly” is interpretative, but we did not remove it because we want to highlight the following text regarding the findings in the UV group.
Also provide interquartile ranges (IQR)
R: Thank you, the IQR data have been added into Table 2.
The paragraph lacks clarity and precision. The authors often use phrases like „respecting IgG“ or “as for IgA“ instead of academic transitions like „in terms of“ or „regarding“. Some sentences are too long or use similar information. Please shortened for better reading flow. Also, present the findings in a more structural manner. The authors should also check the grammar (e.g. subject-verb agreement). The mention of the Pearson χ² test is redundant, as it was already described in the methods section. Instead, the authors should relate the results to the research question, e.g. no significant association was found between the symptoms and systemic and mucosal humoral immune response in HCW.
R: Ok, the paragraph has been modified and shortened; some grammatical connectors were changed to more academic ones, the grammar has been revised, and the mention of the Pearson χ² test was removed.
I don‘t really understand the introduction to the discussion. What does “unclear issues regarding risks and manegment…” have to do with the aim of the study, which is to compare the systemic and mucosal immune response in HCW?
R: These lines have been deleted.
The authors often describe their results and those of others in very long, complex and convoluted sentences. In order to improve the flow of the discussion and comprehensibility, I recommend a stylistic and grammatical revision of the discussion.
R: Thank you, the manuscript has been submitted for revision to a professional stylistic and grammatical editing service.
Nevertheless = However
R: The word has been corrected.
The sentence is correct but it lacks more clarity and better style. I suggest to rewrite the sentence and shorten it where possible.
R: Ok, the paragraph has been modified according to your recommendation.
Avoid the explanation of the groups, as this has already been explained in the
methods section
R: Ok, the explanation of the groups has been deleted.
„… the NE group“ = „.. that the NE group
R: Thank you, the text has been modified.
Start a new sentence: „…NCP; thus,…“
Start a new sentence: „…COVID-19 disease; therefore,…“
R: The changes have been made.
„as previously reported, time post-infection“ - Where in the text is it stated?
R: The sentence has been modified (line 348-351).
Why was this information not recorded? What were the reasons, because there are clear differences in the immune response between the number of vaccine doses and the type of vaccination administered. This is a major limitation in this study and limits the relevance of the results considerably, especially as the time between the last vaccination dose or infection was not recorded
R: Unfortunately, because some of the information was obtained through a questionnaire administered directly to the HCWs, some items containing important information were not answered; we assume this was due to the burnout experienced by the HCWs during that period. This is explained in the text as a limitation of the present study. (lines 351-355)
The conclusion fails to elaborate on the practical consequences of the presented
results. The authors do not really address why the comparison of the systemic and mucosal immune response (study aim) is relevant in practice. The conclusion lacks a connection to how these serol. profiles might inform clinical decision-making, occupational health strategies, or vaccination policies.
R: Thank you very much for your suggestion. The Conclusion has been modified according your recommendations. (lines 371-380).
Reviewer 2 Report
Comments and Suggestions for Authors
This manuscript explores humoral immune responses to SARS-CoV-2 in healthcare workers (HCWs), stratified by immunoglobulin isotype (IgG, IgA, and IgM) and sample source (serum and saliva). This is a valuable contribution to the literature, as most seroprevalence studies focus exclusively on IgG in serum. The inclusion of mucosal (salivary) IgA adds a novel angle, especially in a population with high-risk occupational exposure.
The manuscript is clearly structured and presents useful data. However, several important factual, methodological, and reporting issues should be addressed before publication.
Comments:
1. Incorrect epidemiological statement (Line 46):
The manuscript states that COVID-19 was declared a "pandemic" on 30 January 2020, which is inaccurate.
30 January 2020: Declared a Public Health Emergency of International Concern (PHEIC)
11 March 2020: Declared a Pandemic
References:
- PHEIC declaration https://www.who.int/news/item/30-01-2020-statement-on-the-second-meeting-of-the-international-health-regulations-(2005)-emergency-committee-regarding-the-outbreak-of-novel-coronavirus-(2019-ncov)
- Pandemic declaration https://www.who.int/director-general/speeches/detail/who-director-general-s-opening-remarks-at-the-media-briefing-on-covid-19---11-march-2020
2. COVID-19 vaccination detail lacking (Table 1 & Methods):
Since this study evaluates humoral responses, a detailed vaccination history is essential for clarification.
Consider adding the following information (either in Table 1 or Supplementary Materials):
-Total number of doses (e.g., 1, 2, 3+)
-Type of vaccine(s) received (e.g., BNT162b2, mRNA-1273, ChAdOx1-S, Sputnik V)
-Interval between most recent dose and sampling (e.g., 14–30 days, 31–90 days, >90 days)
These factors significantly affect immune responses and will support the interpretation of antibody titer differences, even if you did not subgroup the humoral immunity comparison.
3. Clarification on Sputnik V (Line 70):
Please clarify the full vaccine name and source as "Sputnik V (Gamaleya)" to be consistent with how other vaccines are referenced.
4. GraphPad software (Line 138):
Specify the exact version number of GraphPad Prism (e.g., “version 6.07”) to ensure reproducibility.
The current citation lists GraphPad as a web address. As this is a desktop program, use the conventional format: “GraphPad Prism v6.07 (GraphPad Software, San Diego, CA, USA)”
Furthermore, version 6 is outdated. Consider checking and confirming it.
5. Comorbidities reporting (Line 98):
While “comorbidities” are mentioned, only low weight and obesity are reported in Table 1.
Consider clarifying data collected for other common conditions (e.g., diabetes, hypertension, respiratory diseases).
If not, clarify in the Limitations section.
6. Figure 2, visual harmonisation:
Currently, each panel has a different Y-axis scale, which makes direct visual comparison between groups difficult.
Consider normalising the y-axis throughout subfigures to the same scale to make it comparable for each Ig isotype and target by the naked eye.
7. Table 1:
You can use multiple comparisons of three groups by using Fisher's exact or Chi-squared (categorical), and ANOVA or Kruskal-Wallis H (continuous) and report the p-value in an additional table column.
Errors.
1. Table 1, footnote annotation (“SD²”):
The superscript “²” placed on “SD” appears to imply “SD squared” (i.e., variance), which is incorrect.
Additionally, the footnotes are out of order: “Obesity” should be ¹, and “SD” should be ².
Suggest correcting to: Obesity¹, SD².
2. Line 67–68, “intradermally”:
Consider verifying the claim that COVID-19 vaccines were administered intradermally.
Almost all WHO or each national FDA-authorised vaccines (including BNT162b2, mRNA-1273, ChAdOx1-S, Sputnik V) were administered via intramuscular injection.
Unless supported by documentation of off-label use or specific local policy, revise to “intramuscularly”.
Author Response
Reviewer 2
Thank you very much for your comments and suggestions, which have greatly helped to improve our manuscript.
Comments:
- Incorrect epidemiological statement (Line 46):
The manuscript states that COVID-19 was declared a "pandemic" on 30 January 2020, which is inaccurate.
30 January 2020: Declared a Public Health Emergency of International Concern (PHEIC)
11 March 2020: Declared a Pandemic
References:
- PHEIC declaration https://www.who.int/news/item/30-01-2020-statement-on-the-second-meeting-of-the-international-health-regulations-(2005)-emergency-committee-regarding-the-outbreak-of-novel-coronavirus-(2019-ncov)
- Pandemic declaration https://www.who.int/director-general/speeches/detail/who-director-general-s-opening-remarks-at-the-media-briefing-on-covid-19---11-march-2020
R: Thank you for your comment. However, heeding the suggestions of other reviewers, the Introduction was modified and that paragraph was deleted in this new version of our manuscript.
COVID-19 vaccination detail lacking (Table 1 & Methods):
Since this study evaluates humoral responses, a detailed vaccination history is essential for clarification.
Consider adding the following information (either in Table 1 or Supplementary Materials):
-Total number of doses (e.g., 1, 2, 3+)
-Type of vaccine(s) received (e.g., BNT162b2, mRNA-1273, ChAdOx1-S, Sputnik V)
-Interval between most recent dose and sampling (e.g., 14–30 days, 31–90 days, >90 days)
These factors significantly affect immune responses and will support the interpretation of antibody titer differences, even if you did not subgroup the humoral immunity comparison.
R: Thank you very much, the COVID-19 vaccination details have been added (Table 1 & Results-lines 183-186).
Clarification on Sputnik V (Line 70):Please clarify the full vaccine name and source as "Sputnik V (Gamaleya)" to be consistent with how other vaccines are referenced.
R: Ok, thank you. The information has been added.
- GraphPad software (Line 138):Specify the exact version number of GraphPad Prism (e.g., “version 6.07”) to ensure reproducibility.
The current citation lists GraphPad as a web address. As this is a desktop program, use the conventional format: “GraphPad Prism v6.07 (GraphPad Software, San Diego, CA, USA)”
Furthermore, version 6 is outdated. Consider checking and confirming it.
R: Thank you, the correct information has been added.
- Comorbidities reporting (Line 98): While “comorbidities” are mentioned, only low weight and obesity are reported in Table 1.
Consider clarifying data collected for other common conditions (e.g., diabetes, hypertension, respiratory diseases). If not, clarify in the Limitations section.
R: The information has been added in Table 1.
Figure 2, visual harmonisation: Currently, each panel has a different Y-axis scale, which makes direct visual comparison between groups difficult.
Consider normalising the y-axis throughout subfigures to the same scale to make it comparable for each Ig isotype and target by the naked eye.
R: Thank you, your suggestion has been implemented in Figure 2.
Table 1:
You can use multiple comparisons of three groups by using Fisher's exact or Chi-squared (categorical), and ANOVA or Kruskal-Wallis H (continuous) and report the p-value in an additional table column.
R: As suggested, an ANOVA was performed for continuous data and Pearson's χ test or Fisher's test for categorical data. This has been included in the methodology section of the statistical analysis. The p-value was included in Table 1 in three columns, comparing HI vs. VI, HI vs. UV, and VI vs. UV.
Errors
- Table 1, footnote annotation (“SD²”):
The superscript “²” placed on “SD” appears to imply “SD squared” (i.e., variance), which is incorrect.
Additionally, the footnotes are out of order: “Obesity” should be ¹, and “SD” should be ².
Suggest correcting to: Obesity¹, SD².
R: Thank you, we have corrected the superscript.
Line 67–68, “intradermally”:
Consider verifying the claim that COVID-19 vaccines were administered intradermally.
Almost all WHO or each national FDA-authorised vaccines (including BNT162b2, mRNA-1273, ChAdOx1-S, Sputnik V) were administered via intramuscular injection.
Unless supported by documentation of off-label use or specific local policy, revise to “intramuscularly”.
R: We apologize; we have changed “intradermally” to “via intramuscular” (line 85).
Reviewer 3 Report
Comments and Suggestions for Authors
Article: Titers of IgG, IgM and IgA against SARS-CoV-2 in Healthcare Workers from a General Hospital in Mexico City
Authors: Nelly Raquel Gonzalez-Arenas, Mariana Dinazar Chavez-Vargas, Hector Prado-Calleros, Juan Pablo Ramirez Hinojosa, Fernando Martinez-Hernandez, Angelica Olivo-Diaz, Pablo Maravilla, Mirza Romero-Valdovinos and Guillermina Avila-Ramirez
The introduction is excessively long and includes general background on the pandemic that is already well known to the scientific community. It should focus on what is truly relevant for the study: occupational exposure of healthcare workers, the humoral immune response to different forms of SARS‑CoV‑2 exposure, and the importance of serological surveillance in this group. Several grammatical and spelling errors (“in the begging”, among others) require careful language editing.
In the study design, the definitions of the HI, VI and NE groups are appropriate, but there is ambiguity regarding exclusion of prior infections. For instance, stating that VI participants never had COVID‑19 conflicts with the detection of NCP antibodies, which can indicate unrecognized asymptomatic infection. Clarify how previous infection was ruled out in the NE and VI groups and discuss limitations of self‑report and a single RT‑PCR test. It would also help to specify when vaccination began in the country and hospital to contextualize group formation.
Laboratory methods are described in adequate technical detail, but the lack of cross‑validation against commercial or standardized assays limits comparability with other series. Provide fuller internal validation of the in‑house ELISA, including intra‑ and inter‑assay reproducibility and sensitivity/specificity confidence intervals. Using 2009 samples as negative controls raises questions about representativeness given possible cross‑reactivity with endemic coronaviruses; add a stronger justification.
Results are well organized in tables and figures, but there is redundancy between narrative and tables. Streamline the text to highlight key comparisons and statistically significant differences. The higher mucosal IgA positivity in the NE group is one of the most interesting findings and deserves deeper discussion of practical implications. Statistical analysis seems sound, yet the absence of association between antibody titers and symptoms should be explored further, especially considering the small number of symptomatic participants in some subgroups.
The discussion is repetitive and occasionally strays from the study objective. Literature comparison is extensive but lacks focus; large sections revisit basic viral immunology and ELISA fundamentals, diluting critical analysis of the authors’ own data. The hypothesis of asymptomatic infection in the VI group should be presented more coherently, including possible cross‑reaction with seasonal coronaviruses. The section on salivary testing is relevant but could probe more deeply into the clinical usefulness of IgA detection in occupational surveillance.
Limitations are acknowledged but could be expanded to mention selection bias (volunteer healthcare workers), lack of longitudinal follow‑up, and limited sample size for some associations. Absence of information on time since last vaccine dose is a key methodological limitation that must be noted.
The conclusion is clear but not sufficiently critical. Add comments on practical applications in clinical and public‑health settings and on future research needs (e.g., longitudinal studies stratified by vaccine type and dose number). Finally, review formal and language issues throughout, standardize acronyms and style (e.g., consistent use of “anti‑RBD” rather than “vs RBD”), and fix truncated phrases in tables.
Author Response
Reviewer 3
Thank you very much for your comments and suggestions, which allowed us to improve our manuscript.
The introduction is excessively long and includes general background on the pandemic that is already well known to the scientific community. It should focus on what is truly relevant for the study: occupational exposure of healthcare workers, the humoral immune response to different forms of SARS CoV 2 exposure, and the importance of serological surveillance in this group. Several grammatical and spelling errors (“in the begging”, among others) require careful language editing.
R: The introduction has been shortened and modified, according to the recommendations of all reviewers, and submitted for revision to a professional stylistic and grammatical editing service.
In the study design, the definitions of the HI, VI and NE groups are appropriate, but there is ambiguity regarding exclusion of prior infections. For instance, stating that VI participants never had COVID 19 conflicts with the detection of NCP antibodies, which can indicate unrecognized asymptomatic infection. Clarify how previous infection was ruled out in the NE and VI groups and discuss limitations of self report and a single RT PCR test. It would also help to specify when vaccination began in the country and hospital to contextualize group formation.
R: In the new version of our manuscript, we discuss the probability of cross-infection with other seasonal coronaviruses (lines 269-273). In addition, information about vaccination has been added (Table 1 and lines 183-186).
Laboratory methods are described in adequate technical detail, but the lack of cross validation against commercial or standardized assays limits comparability with other series. Provide fuller internal validation of the in house ELISA, including intra and inter assay reproducibility and sensitivity/specificity confidence intervals. Using 2009 samples as negative controls raises questions about representativeness given possible cross reactivity with endemic coronaviruses; add a stronger justification.
R: Internal validation of the in-house ELISA, including intra- and inter-assay reproducibility and sensitivity/specificity confidence intervals, is mentioned in the revised version of our manuscript (lines 204-213, and Supplementary Table 1).
Results are well organized in tables and figures, but there is redundancy between narrative and tables. Streamline the text to highlight key comparisons and statistically significant differences. The higher mucosal IgA positivity in the NE group is one of the most interesting findings and deserves deeper discussion of practical implications. Statistical analysis seems sound, yet the absence of association between antibody titers and symptoms should be explored further, especially considering the small number of symptomatic participants in some subgroups.
R: Thank you for your comments. The Results section has been modified according to the suggestions of the reviewers, and a careful stylistic and grammatical revision was carried out to improve the flow of the presentation of the results.
The discussion is repetitive and occasionally strays from the study objective. Literature comparison is extensive but lacks focus; large sections revisit basic viral immunology and ELISA fundamentals, diluting critical analysis of the authors’ own data. The hypothesis of asymptomatic infection in the VI group should be presented more coherently, including possible cross reaction with seasonal coronaviruses. The section on salivary testing is relevant but could probe more deeply into the clinical usefulness of IgA detection in occupational surveillance.
R: The Discussion has been modified following the recommendations of all reviewers. The possible cross reaction with seasonal coronaviruses was analyzed (lines 269-273). Salivary testing is now also discussed in this version of our manuscript (lines 340-347).
Limitations are acknowledged but could be expanded to mention selection bias (volunteer healthcare workers), lack of longitudinal follow up, and limited sample size for some associations. Absence of information on time since last vaccine dose is a key methodological limitation that must be noted.
R: You are right. In this version, we have expanded the limitations of our manuscript to warn readers about possible biases during the recruitment of participants, the sample size, the lack of longitudinal follow-up, and the missing information about the timing of participants’ development of infection and symptoms (lines 348-364).
The conclusion is clear but not sufficiently critical. Add comments on practical applications in clinical and public health settings and on future research needs (e.g., longitudinal studies stratified by vaccine type and dose number). Finally, review formal and language issues throughout, standardize acronyms and style (e.g., consistent use of “anti RBD” rather than “vs RBD”), and fix truncated phrases in tables.
R: Thank you for your comments. The Conclusion has been modified according to your suggestion. A careful review to formalize the use of language and standardize acronyms and style has been performed (Lines 371-380).
Reviewer 4 Report
Comments and Suggestions for Authors
The article entitled “Titers of IgG, IgM, and IgA against SARS-CoV-2 in Healthcare Workers from a General Hospital in Mexico City” is well written. I would like to comment as follows;
This article explored the humoral immune response, specifically IgG, IgM, and IgA antibodies against SARS-CoV-2 in healthcare workers, categorized into three groups: hybrid immunity (HI), vaccine-induced immunity (VI), and natural exposure (NE).
To enhance the quality of the manuscript, revisions should be made following the comments outlined below.
1) The definition of the natural-occupational viral exposure (NE) group is unclear. It is not specified whether this group included individuals with natural infection only. According to Table 1, 100% of the NE group (n=68) were reported as non-infected, raising concerns about the classification. Additionally, this group showed only detection of IgM and IgA against RBD and NCP antigens. It should be clarified whether individuals in the NE group were confirmed COVID-19 cases and, if so, whether they underwent RT-PCR testing during or after recovery.
2) Once again, terminology issues need to be addressed.
2.1) The authors should revise the manuscript to ensure clarity and consistency in the use of terms, particularly when referring to the three study groups: those immunized by vaccination (VI), those with natural-occupational viral exposure or exposition (NE), referred to inconsistently in lines 19, 73 and 87, and those with hybrid immunity (HI) defined as individuals who were both vaccinated and experienced natural infection (lines 74–75).
2.2) It is unclear whether 'natural-occupational viral exposure,' 'exposition,' and 'infection' are intended to describe the same concept or different ones.
2.3) The authors are strongly encouraged to adopt standardized terminology commonly used in COVID-19 literature: HI (Hybrid Immunity – natural infection plus vaccination), VI (Vaccine-Induced Immunity), and NI (Natural Infection) instead of NE.
3) In the Introduction (lines 66–70), the authors stated that in Mexico, doctors initially received the BNT vaccine and were subsequently boosted with various vaccine platforms, including recombinant mRNA vaccines targeting the RBD, viral vector vaccines also targeting the RBD, and whole inactivated vaccines containing both RBD and NCP components. Please clarify how the authors distinguished anti-NCP antibodies derived from vaccination with an inactivated virus (CoronaVac) versus those from natural SARS-CoV-2 infection, particularly in the hybrid immunity (HI) and vaccine-induced immunity (VI) groups. It is important to specify the criteria or methods used in this study to differentiate between these sources of anti-NCP, as both vaccination with inactivated virus and natural infection can elicit NCP-specific antibodies.
3) Reconsider the rearrangement of Table 2 and Figure 1. Due to the MDPI format, the figure and table should be added after their first mention. Please check the format with the author’s guidelines.
4) Check the Figure 2 format
Author Response
Reviewer 4
All co-authors thank you for your comments and suggestions, which greatly helped to improve our manuscript.
To enhance the quality of the manuscript, revisions should be made following the comments outlined below.
1) The definition of the natural-occupational viral exposure (NE) group is unclear. It is not specified whether this group included individuals with natural infection only. According to Table 1, 100% of the NE group (n=68) were reported as non-infected, raising concerns about the classification. Additionally, this group showed only detection of IgM and IgA against RBD and NCP antigens. It should be clarified whether individuals in the NE group were confirmed COVID-19 cases and, if so, whether they underwent RT-PCR testing during or after recovery.
R: We modified the definition of the NE group and changed this acronym according to your recommendation. Now, it reads: “…the unvaccinated group at the beginning of the pandemic (UV) that included those participants who reported having had not COVID-19 and were SARS-CoV-2 negative by RT-qPCR at the time of sampling, and had not been vaccinated, as the vaccine was not yet available, since they were recruited from June to December 2020” (lines 108-111).
2.1) The authors should revise the manuscript to ensure clarity and consistency in the use of terms, particularly when referring to the three study groups: those immunized by vaccination (VI), those with natural-occupational viral exposure or exposition (NE), referred to inconsistently in lines 19, 73 and 87, and those with hybrid immunity (HI) defined as individuals who were both vaccinated and experienced natural infection (lines 74–75).
R: Ok, thank you. The consistency of the participants in the three groups was carefully reviewed throughout the text.
2.2) It is unclear whether 'natural-occupational viral exposure,' 'exposition,' and 'infection' are intended to describe the same concept or different ones.
We cannot use the term "infection" for the UV group (in the first version it was the NE group) because all participants were RT-qPCR negative and although they had some mild symptoms and some of them even had anti-NCP antibodies, we cannot be sure that they were infected and were asymptomatic carriers, which is discussed in lines 269-277.
R: A complication in using the terms "exposure" and "natural-occupational viral exposure" is that they are not exclusive to the UV group, as all participants (regardless of their specialty or type of medical care provided) were on the front lines caring for patients infected with COVID-19. Therefore, we decided to clarify this group as UV: unvaccinated.
2.3) The authors are strongly encouraged to adopt standardized terminology commonly used in COVID-19 literature: HI (Hybrid Immunity – natural infection plus vaccination), VI (Vaccine-Induced Immunity), and NI (Natural Infection) instead of NE.
R: Thank you very much, we have followed your recommendation and adopted standardized terminology commonly used in the COVID-19 literature throughout the manuscript (except for the NE group, which was renamed to the UV group because we consider that it shows a relevant characteristic of this group that is distinctive from the others).
3) In the Introduction (lines 66–70), the authors stated that in Mexico, doctors initially received the BNT vaccine and were subsequently boosted with various vaccine platforms, including recombinant mRNA vaccines targeting the RBD, viral vector vaccines also targeting the RBD, and whole inactivated vaccines containing both RBD and NCP components. Please clarify how the authors distinguished anti-NCP antibodies derived from vaccination with an inactivated virus (CoronaVac) versus those from natural SARS-CoV-2 infection, particularly in the hybrid immunity (HI) and vaccine-induced immunity (VI) groups. It is important to specify the criteria or methods used in this study to differentiate between these sources of anti-NCP, as both vaccination with inactivated virus and natural infection can elicit NCP-specific antibodies.
R: With the new information added in the revised version of our manuscript regarding vaccines and the number of doses administered, we observed that the inactivated virus vaccine (CoronaVac) was not administered in our study population. Therefore, there is no potential bias that the production of anti-NCP antibodies was due to vaccination.
3) Reconsider the rearrangement of Table 2 and Figure 1. Due to the MDPI format, the figure and table should be added after their first mention. Please check the format with the author’s guidelines.
R: Thank you, Table 2 and Figure 1 have been modified accordingly.
Round 2
Reviewer 1 Report
Comments and Suggestions for Authors
Dear authors,
you have implemented most of the suggestions, but after the revision, aspects have been changed that raise further questions. I would like to briefly address these here:
The description of the study population in Table 1 lists ten categories for overweight/obesity. This is very confusing. Why don't the authors simply use the three categories based on BMI: 1) underweight/normal weight, 2) overweight, 3) obese, so that they add up to 100% per group?
The category would then have to be renamed “BMI“ and not „Overweight or Obesity“ as the authors also present data on participants with normal weight. This would provide greater clarity and consistency.
Why was no BMI mean calculated for the UV group (Table1)? The categories below contain absolute and relative data. I also suggest presenting either the means or the group proportions, but not both.
The detailed description of the group differences (lines 192–202) is unnecessary and confusing, as each p-value is listed. The results are already in the table and therefore do not need to be documented in such detail in the text. I also wonder what added value these group differences have in relation to the research question. The presentation of the individual BMI groups is also confusing. Here, too, the question arises: What added value do the authors expect in terms of content?
Author Response
Dear anonymous reviewer 1:
We deeply appreciate your comments and suggestions, all of them have greatly helped to improve our manuscript. Thank you very much.
The description of the study population in Table 1 lists ten categories for overweight/obesity. This is very confusing. Why don't the authors simply use the three categories based on BMI: 1) underweight/normal weight, 2) overweight, 3) obese, so that they add up to 100% per group?
R: Thank you, Table 1 was changed according your suggestion using the common categories based on BMI; now in the new version of our manuscript, they add up to 100% per group.
The category would then have to be renamed “BMI” and not “Overweight or Obesity” as the authors also present data on participants with normal weight. This would provide greater clarity and consistency.
R: We decided to retain commonly used terms such as obesity, overweight, and normal weight, following previous recommendations from one of the reviewers. Therefore, to maintain the clarity and consistency of the text, the terms are clarified in the figure caption of Table 1 based on their IBM.
Why was no BMI mean calculated for the UV group (Table1)? The categories below contain absolute and relative data. I also suggest presenting either the means or the group proportions, but not both.
R: Ok, now the number and percentage of participants with obesity in each group (including UV group) is shown in Table 1. Also we included the mean ± standard deviation of BMI in all groups.
The detailed description of the group differences (lines 192–202) is unnecessary and confusing, as each p-value is listed. The results are already in the table and therefore do not need to be documented in such detail in the text. I also wonder what added value these group differences have in relation to the research question. The presentation of the individual BMI groups is also confusing. Here, too, the question arises: What added value do the authors expect in terms of content?
R: Thank you, the paragraph has been shortened and only a brief overview is referred to in the text (lines 186-188). Although the inclusion of participants' weight may raise more questions and divert readers' attention from the main objective of our study, according to one of the reviewers, the inclusion of this variable may be important for the general description of the study population.
Reviewer 2 Report
Comments and Suggestions for Authors
Thank you for thoroughly addressing the concerns I raised in your previous submission. The changes made have improved the clarity and overall quality of the manuscript.
Author Response
Dear anonymous reviewer 2:
All co-authors thank you for your comments and suggestions, which greatly helped to improve our manuscript.
Reviewer 3 Report
Comments and Suggestions for Authors
Review – Minor Revision (2nd)
The manuscript entitled “Titers of IgG, IgM and IgA against SARS-CoV-2 in Healthcare Workers from a General Hospital in Mexico City” presents a solid methodological design and contributes to the understanding of the immune response among healthcare workers exposed to coronavirus disease (COVID)-19 through different contexts (infection history, vaccination, or absence of exposure). The revised version of the manuscript has incorporated several of the previously suggested improvements, including a more focused introduction, enhanced description of laboratory methods, and a more objective discussion.
Positive aspects of the revised version:
- The introduction was shortened and now focuses on occupational exposure and serological surveillance, aligning more closely with the study’s aim.
- The text is clearer and more fluent, with evident language editing and technical term standardization.
- The definitions of the study groups (HI, VI, NE) are clearer, with a discussion of limitations related to self-reporting and single RT-PCR testing.
- The results section is more concise and avoids redundancy with the tables.
- The discussion now better emphasizes the study's findings, especially the higher mucosal IgA positivity in the NE group.
Minor revision suggestions:
- ELISA validation: Although improvements were made, it is recommended to include confidence intervals (CIs) for sensitivity and specificity, if available, even if based on internal validation.
- Negative controls (2009 samples): A stronger justification for the use of these samples is encouraged, given the possibility of cross-reactivity with seasonal coronaviruses.
- Conclusion: Briefly add practical implications for occupational surveillance and potential use in future screening strategies.
- If possible, increase the resolution of the figures.
Recommendation:
Minor Revision – The manuscript is close to final and only requires a few adjustments, with no need for another full review round.
Author Response
Dear anonymous reviewer 3:
Thank you very much for your comments and suggestions, which greatly helped to improve our manuscript.
ELISA validation: Although improvements were made, it is recommended to include confidence intervals (CIs) for sensitivity and specificity, if available, even if based on internal validation.
R: Thank you for your observations, the confidence intervals (CIs) were calculated and are shown in the new Supplementary Table 1, sheet 1(line 204).
Negative controls (2009 samples): A stronger justification for the use of these samples is encouraged, given the possibility of cross-reactivity with seasonal coronaviruses.
R: Certainly, the possibility of cross-reactivity with seasonal coronaviruses in controls cannot be ruled out, however this limitation is mentioned in the discussion (lines 265-270), where it is stated that in pre-COVID-19 studies in hospitals in Mexico, the identification of seasonal coronaviruses was <0.6% (Wong-Chew et al 2017. Int J Infect Dis. 62:32-38. doi: 10.1016/j.ijid.2017.06.020; Gamiño-Arroyo et al., 2017. Influenza Other Respir Viruses. 11(1):48-56. doi: 10.1111/irv.12414).
Conclusion: Briefly add practical implications for occupational surveillance and potential use in future screening strategies.
R: Thank you, the following text has been added to the conclusion: Furthermore, the epidemiological description of seroprevalence and antibody titers is valuable in providing information on the dynamics of the COVID-19 epidemic, particularly in groups occupationally exposed to the virus, such as healthcare workers (lines 378-381).
If possible, increase the resolution of the figures
R: Ok thank you, quality of figures was improved